# Multilayer, Broadband Infrared Reflectors Based on the Photoinduced Preparation of Cholesteric Liquid Crystal Polymers

**DOI:** 10.3390/molecules28207063

**Published:** 2023-10-12

**Authors:** Yutong Liu, Rui Han, Xiaohui Zhao, Yue Cao, Hui Cao, Yinjie Chen, Zhou Yang, Dong Wang, Wanli He

**Affiliations:** 1School of Materials Science and Engineering, University of Science and Technology Beijing, Beijing 100083, China; lytong99@163.com (Y.L.); hanr047@163.com (R.H.); m202120498@xs.ustb.edu.cn (X.Z.); m202210273@xs.ustb.edu.cn (Y.C.); yangz@ustb.edu.cn (Z.Y.); wangdong@ustb.edu.cn (D.W.); hewanli@mater.ustb.edu.cn (W.H.); 2Beijing Engineering Research Center of Printed Electronics, Beijing Institute of Graphic Communication, Beijing 102600, China

**Keywords:** cholesteric liquid crystal, broadband reflection, pitch gradient, photoinduced diffusion

## Abstract

This paper focuses on preparing broadband reflective liquid crystal films through the diffusion of monofunctional and bifunctional monomers in a photoinduced trilayer system. By combining the hydrophilic and hydrophobic liquid crystal glass surface treatment technologies, the polymer network of polymer-stabilized cholesteric liquid crystal (PSCLC) itself serves as a diffusion channel to form a trilayer cholesteric liquid crystal composite system containing bifunctional monomers, a nematic liquid crystal composite system, and a cholesteric liquid crystal composite system containing monofunctional monomers. Utilizing the difference in the polymerization rates of monofunctional and difunctional polymerizable monomers, the monomers and chiral compounds diffuse relative to each other, so that the liquid crystal pitch exhibits a gradient distribution, and the broadened reflective width can reach up to 1570 nm. There is no doubt that new and improved processes and technologies offer important possibilities for preparing and applying PSCLC films.

## 1. Introduction

Cholesteric liquid crystal (CLC) materials with special electrical and optical properties have tremendous potential for applications and have gradually attracted research interest from scholars in different fields [1]. The optical properties of cholesteric liquid crystals (CLCs) mainly include the birefringence phenomenon, rotational properties, circular dichroism, and selective reflectance [2,3,4]. Selective reflexes are due to a unique spiral structure and Bragg’s Law [5]. However, the selective reflection property of cholesteric liquid crystals is not manifested at all wavelengths of light; the reflected wavelength is determined by λ = nP, and the reflected bandwidth is determined by ∆λ = ∆nP, which are both derived from the Bragg basis formula [6,7]. In this formula, n represents the average refractive index of the cholesteric liquid crystals, P is the pitch of the cholesteric liquid crystal, and ∆n represents the CLC birefringence (∆n = n_e_ − n_o_); λ, P and ∆λ have the same unit of nm, while n and ∆n have dimensionless units; and P = [(HTP) × c]^−1^, while HTP is the helical torsion force of the chiral compound, and c is the chiral compound content. The selective reflection wavelength of cholesteric liquid crystals can vary due to external field stimuli, such as temperature [8,9,10], forces [11,12], light [13,14,15], electricity [16], humidity [17], etc.

Utilizing the above principles, cholesteric phase liquid crystals have been investigated in a number of directions, including broadband reflectors, hyper-reflective films, liquid crystal elastomers, and so on [18,19]. Our research direction is the combination of broadband reflection and a polymer network to prepare polymer-dispersed liquid crystals (PDLCs) and polymer-stabilized liquid crystals (PSLCs). The main dissimilarity between PSLCs and PDLCs is the polymer content, which is usually greater than 10% in PDLCs, resulting in PDLCs exhibiting other outstanding properties, along with excellent mechanical properties [20]. The method of broadband reflection using a light-induced molecular gradient distribution was first proposed by the Dutch scientist Broer [21,22], in whose research the material system of bifunctional chiral liquid crystalline polymerizable monomers, monofunctional liquid crystalline polymerizable monomers, UV-absorbing dyes, and photoinitiators was used for the first time. There have been many reports on this research in recent years. Mitov investigated a small-molecule liquid crystal/bifunctional liquid crystalline polymerizable monomer/photoinitiator system and found that the asymmetric irradiation of the liquid crystal cassettes with weak ultraviolet (UV) light induced a concentration gradient in the polymer network, with the concentration of the polymer network being higher on the side near the UV source and lower on the side far away from the UV source. Consequently, the selective reflectance bandwidth of the polymerized Ch was widened from 80 nm to 220 nm, while under symmetric irradiation (the simultaneous irradiation of both sides of the liquid crystal cassettes with the same intensity of UV), there was no obvious gradient distribution in the polymer network, and the reflective bandwidth was narrower than that under asymmetric irradiation [23]. Differences in reactivity between bifunctional and monofunctional monomers and the UV intensity gradient perpendicular to the film due to the presence of the dye facilitated the formation of the gradient. Through the spin-coating and assembly of two layers of cholesteric photopolymerized monomers/chiral compounds with different pitches, at the same temperature, Sixous obtained a more or less significant diffusion between the two layers due to the concentration gradient leading to the diffusion of the chiral compounds across the film’s thickness, depending on various parameters such as the degree of cross-linking of the two layers, the thickness of the layer, the temperature dependence, and the time at a given temperature [24,25]. Yang synthesized chiral thiol molecules of isosorbitol derivatives with a bipartite thiol functional group (RIS) with a high HTP and successfully prepared PSCLC films with broadband reflective properties using thiol-acrylate click chemistry. The polymerization rate of the click reaction was faster than the free radical polymerization of the acrylic monomer, and the C6M and RIS diffused to the corresponding positions with shorter spacing. A pitch gradient formed in the direction of UV irradiation to realize broadband reflection [26]. Yoon used an optically forced layering method to fabricate single-substrate flexible thermo-responsive CLC films. By varying the chiral dopant content, the transmission spectra of the films were controlled to undergo blue-shift or red-shift phenomena and also exhibited temperature-responsive properties [27]. Albertus’ research team polymerized liquid crystal systems containing mono- and bifunctional compounds through UV irradiation to obtain films capable of broadband reflection and then overlapped broadband reflective films containing chiral compounds with different rotational orientations to successfully prepare infrared reflectors capable of broadband reflection with a transmittance of more than 60% [28]. Schenning created a pitch gradient by pretreating a substrate to form a photoinitiator concentration difference, diffusing it into the coating to cause a polymerization reaction concentration difference [29].

Today, the methods widely used to prepare broadband reflective cholesteric liquid crystal films include the layer-stacking method, heat-induced molecular diffusion method, the HTP value change method for chiral compounds, the electromagnetic-field-induced pitch non-uniform distribution method, and so on. Huang et al. stacked three layers of ChLC polymers with different pitches to create a composite system with reflectance waveforms covering the reflectance range of each layer. The advantages of the single-pitch multilayer stacking method are the simplicity of the preparation process and the controllability of the reflection wavelength center and the reflection wave width range [30]. Mitov proposed a method in which liquid crystal oligomer films of cyclosiloxane side chains with different proportions of chiral and achiral side chain branching are directly stacked, and a pitch gradient is formed between the two films through thermal diffusion after certain heat treatments [31]. Yang et al. prepared azo chiral compounds with photoresponsive properties. UV irradiation was applied to induce the cis–trans isomerization of the azo group, which led to a decrease in the helical twisting force, as well as the red-shifting of the reflection peaks, and the cholesteric-phase reflection peaks changed with the increase in the UV irradiation time. The cis-trans isomerization of azo chiral compounds in different positions progresses to different degrees, resulting in a gradient distribution of helical twisting force and a significant enhancement of reflectivity while realizing broadband reflection [32]. Later, Yang et al. added an anionic chiral ionic liquid containing chiral groups to a ChLC material system. They found that under the action of an applied high-frequency AC electric field, the anions in the ionic liquid move toward the positive electrode, resulting in a higher concentration of chiral groups near the positive electrode and a lower concentration near the negative electrode, which creates a gradient of concentrated chiral compounds. The reflected wavelengths cover the entire visible wavelength band when the applied voltage reaches 40 V [16].

CLC films with broadband reflective properties are attractive and widely demanded in the market, both in scientific research and in practical industrialization. The findings reported in this manuscript are mainly related to the near-infrared region. In real life, the primary application is to attach the films to the windows of buildings to obtain infrared-adjustable smart windows, for which one can effectively adjust the incidence and reflection of infrared light, effectively reduce the indoor temperature in the summer and the frequency of using air conditioners, and realize energy savings and emission reductions [33]. This technology can also be used in military and medical applications as a laser protection film. The wider the broadband of the shielding is, the better the protection film will be for medical and military applications [34].

In our previous study, UV-absorbing dyes and chiral compounds were loaded onto porous ZIF-8 materials and then fixed on glass plates on both sides of liquid crystal cassettes to diffuse relative to each other [35]. Under UV irradiation, the intensity of light was distributed as a gradient, which promoted the relative diffusion of the chiral compounds and the polymerizable monomers to produce a concentration gradient and caused a pitch gradient distribution to achieve the effect of broadband reflection. In this study, we improved the previous single-layer method to form a triple-layer cholesteric liquid crystal composite system, with a transition from a cholesteric phase containing a bifunctional monomer to a nematic phase and then a cholesteric phase containing a monofunctional monomer, by combining the glass surface treatment technology of the hydrophilic and hydrophobic liquid crystal methods and using the polymer network of the PSCLC itself as the diffusion channel. Using the polymerization rate difference between the mono- and bifunctional polymerizable monomers, different conditions were further controlled to ensure the relative diffusion of the mono- and bifunctional monomers and chiral compounds to form a pitch gradient and to broaden the reflective bandwidth. In this paper, the influences of the content of polymerizable monomers, UV irradiation intensity, polymerization time, and the content of chiral compounds in the liquid crystal composite system on the reflected bandwidths of the prepared cholesteric liquid crystal films are investigated, and the formation of a pitch gradient is demonstrated.

## 2. Results

### 2.1. Mechanisms of the Broadband Reflection Phenomenon

The diffusion of dye is mainly utilized to cause a non-uniform distribution of UV intensity, and the polymerization of bis-acrylate monomers occurs after their movement to the stronger side of the UV range to form an inhomogeneous polymer network, while at the same time, the chiral compounds and the monoacrylate diffuse in the opposite direction [36,37,38]. According to diffusion kinetics, ideally, substances generally diffuse from the high-concentration to the low-concentration side. As polymerization proceeds, less unpolymerized bisacrylate monomer remains on the stronger UV side, which promotes diffusion. However, the reflective broadband of the film is related to not only the diffusion rate but also the polymerization rate. With moderate diffusion and polymerization rates, chiral compounds can form a good concentration gradient, i.e., different concentrations at different locations in the cross-sectional direction within a small range, and the pitch can also form a gradient distribution state according to P = [(HTP) × c]^−1^, thus realizing the reflection broadband phenomenon as characterized via transmission spectroscopy.

The rate of polymerization is related to the following equation under steady-state conditions in a dilution system [39,40]:$$Rp=Kp\left[M\right]{\left\{\emptyset I_{0}\left(1-\left.10^{-\varepsilon \left[In\right]d}\right)\right.\right.\left./Kt\right\}}^{1/2}$$
where Rp represents the rate of polymerization, Kp and Kt represent the rate constants of propagation and termination, [M] represents the monomer concentration, Φ represents the overall quantum efficiency for dissociation and initiation, ε represents the molar extinction coefficient of the photoinitiator, [In] represents the concentration of the photoinitiator, I_0_ represents the intensity of the incident light, and d represents the thickness of the sample. The polymerization and diffusion rates are controlled to adjust the gradient distribution of chiral compounds in the cross-sectional direction of the film in order to achieve the broadband reflection of the film.

Figure 1 shows the mechanism of the reflection band’s broadening in this study. The CLC composite system containing the bifunctional monomer C6M was added into the liquid crystal cassette of the near-light layer, and the CLC composite system containing the monofunctional monomer HCM021 was added into the liquid crystal cassette of the far-light layer. In Figure 1a, the pitch of the CLC composite system is uniform before pre-polymerization. 

In Figure 1b, the pre-polymerization of the near-light and far-light layers, respectively, serves to form a certain polymer network, fixing the formed CLC pitch so that the film can be separated from the liquid crystal cassette as relatively intact. Due to the presence of UV-absorbing dyes in the system, a certain light intensity gradient is formed in the direction of the box thickness during pre-polymerization, which results in a gradient change in the polymerization rate of the polymerizable monomers in the direction of the box thickness, so that the unpolymerized C6M in the near-light layer migrates from the far side of the light to the near side of the light, and at the same time, the chiral compounds migrate relative to the monomers, which generates the gradient distribution. The polymerizable chiral molecule HCM006 in the far-light layer migrates to the near-light side to produce a gradient distribution, and according to Eq. P = [(HTP) × c]^−1^, the difference in the concentration of chiral dopants leads to the formation of the pitch gradient of the liquid crystal composite system, but the pitch gradient is not formed completely due to the short polymerization time. 

The liquid crystal cassettes of the separated near and far optical layers were reassembled into new liquid crystal cassettes with a thickness of 20 μm and infused into the CLC composite system containing both the polymerizable monomers C6M and HCM021 in order to form a trilayer liquid crystal film, as shown in Figure 1c. The liquid crystal cassettes were placed with the near-light layer facing up and the far-light layer facing down and diffused at 40 °C for 5 min. Using the polymer networks of the near- and far-light layers themselves as diffusion channels, the dye molecules and chiral molecules diffused into the intermediate layer because of the concentration difference between them, forming a new cholesteric phase in the intermediate layer. Figure 1d demonstrates that, along with the time change in UV irradiation curing, the UV-absorbing dyes in the near-light layer system diffuse to the middle layer system, forming a UV intensity gradient in the middle layer, and because of differences in the reactivity of mono- and bifunctional polymerizable monomers, the rate of C6M polymerization is faster than that of HCM021; then, the migration of the unpolymerized C6M in the middle layer to the near-light layer prompts the migration of R5011 in the near-light layer towards the middle layer, which further expands the upper-layer pitch gradient. The HCM021 in the middle layer migrates to the far light layer, and a light intensity gradient is formed in the middle layer. The HCM006 in the far light layer migrates to the middle layer, further expanding the pitch gradient in the lower layer and ultimately forming a pitch gradient from the small pitch in the middle to the large pitch on both sides. After UV polymerization, the pitch gradient distribution becomes fixed by forming a stable polymer network.

### 2.2. The Effect of the Content of Polymerizable Monomers in the Middle Layer on the Broadband Reflectance of the Samples

Regarding the samples of group Ac, the mass fractions of the polymerizable monomers C6M and HCM021, infused into the middle layer, were set to be 4 wt%, 6 wt%, 7 wt%, 8 wt%, and 9 wt%, respectively, and the rest of the sample grouping ratios are shown in Table 1. The samples obtained were named Ac1, Ac2, Ac3, Ac4, and Ac5, respectively. All groups of samples were polymerized at a light intensity of 1.0 mW/cm^2^ for 25 min. The transmission spectra of the samples of group Ac are shown in Figure 2, where A and c denote the UV transmission spectra of the upper and lower samples after individual polymerization, respectively. As seen in Figure 2, the Δλ of samples Ac1 to Ac3 shows an increasing trend with the increase in the mass fraction of the polymerizable monomers; the Δλ of samples Ac3 to Ac5 shows a decreasing trend with the increase in the mass fraction of the monomers. When the C6M and HCM021 mass fractions amount to 7 wt%, the maximum Δλ is obtained, corresponding to 1570 nm. This trend of increasing and then decreasing is the result of the simultaneous influence of the polymerization rate and molecular diffusion when the monomer mass fraction in the liquid crystal composite system is changed. Following the formation of the trilayer system, the polymerization rate is lower when there is less C6M and HCM021 in the system, and the amount of unpolymerized C6M in the upper and middle layers that diffuses to the near-light side is less, and less R5011 chiral molecules in the upper layer diffuse in the direction of the middle layer, which results in the inability of the upper layer to form a larger pitch. Moreover, the diffusion of HCM021 from the middle layer to the lower layer is also less significant, which, in turn, contributes to the inefficiency of the diffusion of HCM006 to the middle layer; thus, the lower layer is unable to form a larger pitch gradient. Meanwhile, the polymer network formed through polymerization is not dense enough to have a strong anchoring effect on the pitch gradient. As the content increases, the C6M polymerization rate and diffusion rate accelerate, the diffusion of R5011 from the upper layer to the middle layer increases, and the pitch gradient formed in the upper layer increases. It also induces an increase in the diffusion efficiency of HCM021 in the middle layer, causing it to diffuse into the lower layer, which, in turn, increases the efficiency of HCM006 in diffusing into the middle layer. Therefore, a larger pitch gradient is formed in the lower layer. The polymer network formed through polymerization is denser and acts as an anchor for the pitch distribution. Thus, a broadening of the Δλ to the maximum value of the reflected bandwidth is realized. When the content continues to increase, the polymerization rate of the monomer is too fast, being greater than the diffusion rate of the chiral molecules, thus hindering the formation of the pitch gradient. Simultaneously, the polymer network formed through polymerization is too dense, which plays a hindering role in the diffusion of the unpolymerized monomer and chiral molecules, affecting the gradient distribution of pitch and thereby gradually reducing the Δλ.

### 2.3. The Effect of the R5011 Content on the Reflected Bandwidth of the Samples

The content of the chiral compound R5011 was set to be 1.00 wt%, 1.05 wt%, 1.10 wt%, 1.15 wt%, and 1.20 wt% for the samples of the Ag group, and the UV intensity was 1.0 mW/cm^2^. Polymerization was carried out for 25 min, and the samples were named as Ag1, Ag2, Ag3, Ag4, and Ag5, respectively. The transmission spectra of the Ag group samples are shown in Figure 3, in which A and g denote the UV transmission spectra of the upper and lower samples after the polymerization of the upper and lower layers, respectively.

It can be seen in Figure 3 that the Δλ increases for sample Ag3 compared to Ag1 and Ag2, whereas, the Δλ of samples Ag3 to Ag5 shows a decreasing trend with the increasing R5011 content. When the R5011 content is 1.10 wt%, this corresponds to a maximum Δλ of 1570 nm.

The pitch of the CLC is inversely proportional to the chiral dopant content according to the equation P = [(HTP) × c]^−1^; the central reflection wavelength is proportional to the pitch according to the equation λ = nP. The concentration of chiral molecules in the liquid crystal composite system increases, the pitch decreases, and the reflected bandwidth decreases. Therefore, as the R5011 content becomes higher, the center of reflection gradually blueshifts, and at the same time, it has the effect of narrowing the pitch. However, when secondary polymerization occurs, the Δλ can be broadened again, as the upper R5011 and UV-327 diffuse into the middle nematic phase to form a new cholesteric phase, resulting in a larger pitch gradient distribution. When the content of R5011 is less than 1.10 wt%, the widening effect of the Δλ formed through diffusion is greater than the effect of the narrowing pitch due to the increase in the chiral content, and the Δλ shows a gradual increase as a whole, or vice versa when the content of R5011 continues to increase.

### 2.4. The Effect of Polymerization Time on the Reflection Bandwidth of the Samples

As for the Rw group of samples, the control upper, middle and lower components were as shown in Table 1, and the UV intensity was 1.0 mW/cm^2^. The UV irradiation time was set to 0 min, 10 min, 20 min, 25 min, 30 min, and 35 min, and the samples were named as Rw1 (unpolymerized), Rw2, Rw3, Rw4, Rw5, and Rw6, respectively. The transmission spectra of the samples in group Rw are shown in Figure 4. Intuitively, it can be seen that the Δλ of samples Rw1 to Rw5 shows an increasing trend with the increasing polymerization time; samples Rw5 and Rw6 continue to increase with the polymerization time compared to Rw4, but the Δλ tends to stabilize. The cause is that when the polymerization time is short, the relative diffusion of the monomer molecules is incomplete, and the concentration gradient of the chiral molecules gradually formed due to the diffusion of the molecules is still in the process of expanding; thus, the pitch gradient formed expands, and Δλ continues to increase. After 25 min of polymerization, when the polymerization time continues to increase, the Δλ effectively no longer increases, and it can be assumed that, effectively, the polymerizable monomers have been polymerized completely within the error tolerance, as the polymer network has a completely fixed pitch gradient; thus, the Δλ no longer broadens. Therefore, it can be assumed that the film has been polymerized when the UV irradiation time is 25 min.

### 2.5. The Effect of UV Intensity on the Reflection Bandwidth of the Samples

Regarding the samples of group Dw, the samples were polymerized under UV light of different intensities for 25 min to obtain samples named Dw1, Dw2, Dw3, Dw4, and Dw5, respectively. The corresponding UV intensities were 0.2 mW/cm^2^, 0.5 mW/cm^2^, 1.0 mW/cm^2^, 1.5 mW/cm^2^, 2.0 mW/cm^2^, respectively. The transmission spectra of the samples of group Dw are shown in Figure 5.

It can be intuitively observed that sample Dw2 increases in UV intensity compared to Dw1, and the Δλ also increases from 1324 nm to 1570 nm. Meanwhile, the Δλ of samples Dw2 to Dw5 shows a decreasing trend along with the increase in UV intensity, and it further decreases down to 1142 nm. The maximum Δλ is obtained when the UV intensity is 1.0 mW/cm^2^.

This trend of increasing and then decreasing is a result of the fact that the polymerization rate and molecular diffusion are both affected when the UV light intensity changes. The reflection bandwidth of the Dw1 sample is reduced compared to the Dw2 sample because the slower rate of monomer polymerization makes the diffusion insufficient, and the pitch gradient is not fully formed in the same amount of time. When the UV intensity is increased from 1.0 mW/cm^2^ to 2.5 mW/cm^2^, the polymerization rate of C6M as well as HCM021 is too fast, and the diffusion of the monomers and chiral molecules is no longer dominant. Thus, the concentration gradient formed through the migration of R5011 decreases, preventing the formation of the pitch gradient. In the meantime, the polymerization of the polymerizable chiral HCM006 is also accelerated to the detriment of its own diffusion into the mesosphere. The polymer network formed is too dense, which plays a hindering role in the diffusion of unpolymerized monomer and chiral molecules, and the relative diffusion of chiral molecules is also hindered, which affects the gradient distribution of the pitch, thus gradually decreasing the Δλ.

### 2.6. Optimal Sample Comparison and Liquid Crystal Film Cross-Section SEM Morphology

According to the optimal conditions determined in the abovementioned experiments, the sample Ag3 was prepared and set to polymerize for 5 min at a pre-polymerization UV intensity of 0.5 mW/cm^2^, and after diffusion for 5 min on a hot bench at 40 °C, the sample was polymerized for 25 min in a secondary polymerization with the UV intensity of 1.0 mW/cm^2^. Figure 6a shows the transmission spectra of Ag3 before and after polymerization. In Figure 6a, it can be seen that the reflection bandwidth is 1570 nm after the completion of Ag3 polymerization of the sample, and it can clearly be seen that the Δλ is significantly increased, which proves that the preparation of broadband reflective CLC films using the photoinduced trilayer mono–bifunctional monomer diffusion method allows one to achieve a significant broadening of the Δλ.

## 3. Experimental

### 3.1. Materials

The nematic liquid crystal (SLC-1717, n_e_ = 1.720, n_o_ = 1.519, Δn = (n_e_ − n_o_) = 0.201, Shijiazhuang Chengzhi Yonghua Display Material Co., Ltd., Shijiazhuang, China); the chiral dopant (R5011, Shijiazhuang Chengzhi Yonghua Display Material Co., Ltd. Shijiazhuang, China); the free radical photoinitiator (IRG651, Aladdin Co., Ltd. Shanghai, China); the UV absorption dye (UV-327, Annaiji Chemical Reagent Co., Ltd. Shanghai, China)); the UV radical polymerizable monomer (C6M, Beijing Kexin Jingyuan Technology and Trade Co., Ltd. Beijing, China)); 3-(Methacryloxy) propyltrimethoxysilane (KH-570, silane coupling agent, J&K Scientific Ltd. Beijing, China); 1H, 1H, 2H, and 2H-perfluorodecyltriethoxysilane (J&K Scientific Ltd. Beijing, China); liquid crystal polymerizable monomer HCM021 (Jiangsu Hecheng Co., Ltd. Jiangsu, China); and chiral additive HCM006 (Jiangsu Hecheng Co., Ltd. Jiangsu, China) were used in the study. The chemical structure of the above materials is shown in Figure 7.

### 3.2. Measurements

The transmission spectra of the samples were measured using an ultraviolet–visible–near-infrared spectrophotometer (JASCO-V570, Jasco Co., Ltd., Tsukuba, Japan), and the transmittance of the blank cells was standardized to 100%. In general, Δλ is defined as the bandwidth at half the height of the transmitted light peak. The morphology of the pitch gradient distribution on the cross-section of the films was studied using scanning electron microscopy (SEM, Zeiss-SUPRA55, Jeol Co., Ltd., Tokyo, Japan). A Fourier transform infrared spectrometer (FTIR, Spectrum One, Perkin Elmer Co., Ltd., Waltham, MA, USA) was utilized to analyze the infrared spectra of the samples.

### 3.3. Preparation of Samples and Cells

Functionalization of the glass: Take the cleaned glass sheet and rotary coat silane coupling agent KH-570 (γ-methacryloyloxypropyl) (1 vol% solution in a water–methanol mixture) or polytetrafluoroethylene solution (1 vol% solution in ethanol) placed on the surface of the glass substrate, put it on the rotary coater, set the rotational speed to 1200 rpm for 60 s, and then heat the rotary-coated glasses in a muffle furnace at 150 °C for 20 min so that the solvents volatilize completely.

Assembling the liquid crystal cassettes: Take two pieces of glass with different functionalizations, place the treated sides facing each other, use 20 μm thick polyethylene terephthalate (PET) films as spacer pads and stagger the top and bottom by approximately 0.5 cm, and seal the two sides with 502 strong adhesives.

Using an electronic scale, weigh each experimental material according to the group allocation ratios shown in Table 1 and add these materials to individually numbered centrifuge tubes. Mix the centrifuge tubes with dichloromethane and sonicate for 10 min after shaking and stirring; then, clarify the solution clarified without precipitation. Transfer the samples to brown glass sample bottles and place them in an oven to dry the solvent. Dip the end of a straightened crankpin into the sample and apply the sample to the gap where the two glass substrates are laminated to penetrate the liquid crystal cassettes under a capillary action. Adjust the light intensity of the 365 nm UV light source, set the pre-polymerization time to 5 min and the light intensity to 0.5 mW/cm^2^, and place the perfused liquid crystal cassettes on a 40 °C hot stage for pre-polymerization. After peeling off the hydrophobic liquid crystal layer, leave the liquid crystal film with the gradient distribution of polymer network on the glass of the pro-liquid crystal layer, and place a 20 μm thick PET film in the middle of the upper and lower layers to reconstitute the new liquid crystal cassette. Infuse the nematic liquid crystals into the middle layer of the new liquid crystal cassettes, which are first diffused on a hot bench at 40 °C for 5 min and then subjected to the curing reaction under 365 nm UV light. The whole experimental process needs to be protected from light to avoid the early polymerization of the liquid crystal composite system. The operational procedure for sample preparation is shown in Figure 8.

## 4. Conclusions

In this paper, a three-layer system, that is, a liquid crystal composite system containing bifunctional monomer cholesteric phase to nematic phase to liquid crystal composite system containing monofunctional monomer cholesteric phase, was designed to prepare liquid crystal cassettes transitioning from a cholesteric phase to a nematic phase and then to a cholesteric phase by combining the glass surface treatment technique of the hydrophilic to hydrophobic liquid crystal methods. Utilizing the polymer network of the PSCLC itself as the diffusion channel, due to the difference in the polymerization rates of mono- and bifunctional polymerizable monomers, as well as the competition between polymerization and diffusion, the pitch gradient was formed in the liquid crystal films with a gradual increase in the pitch from the middle to both sides, realizing the broadband reflection of the PSCLC films. We further controlled the infused liquid crystalline polymerizable monomer content, the content of chiral molecules, the UV light intensity and the polymerization time to modulate the variation in the reflected bandwidth. Experimentally, we determined that the Δλ of PSCLC films can reach a maximum of 1570 nm, realizing a significant broadening of the Δλ. Meanwhile, the cross-sectional morphology of the cholesteric liquid crystal films of the samples after the polymerization was completed indirectly confirms that the broadening of the Δλ is caused by the distribution of the pitch gradient. The characteristics of broadband reflective cholesteric films in the IR region could allow them to be widely used in the environmental protection features of architectural glass, military shielding convenience, laser protection, etc. They are materials with an excellent performance and have a wide range of market prospects.

## Figures and Tables

**Figure 1 molecules-28-07063-f001:**
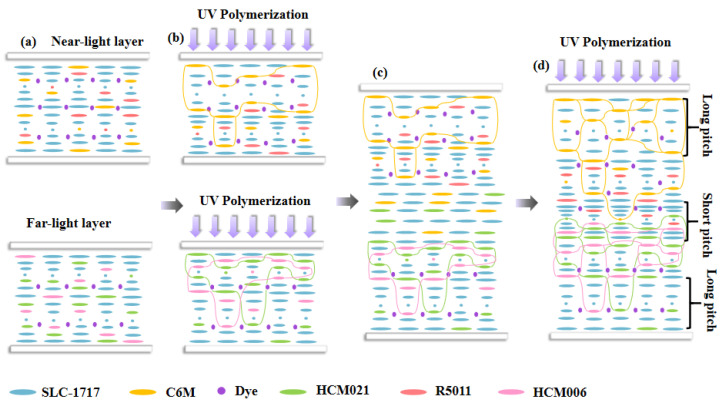
Schematic diagram of the mechanism for broadening the reflected bandwidth using the chiral compound diffusion method. Top and bottom monolayer films before (**a**) and after (**b**) polymerization. Recombinant liquid crystal cassettes before (**c**) and after (**d**) polymerization.

**Figure 2 molecules-28-07063-f002:**
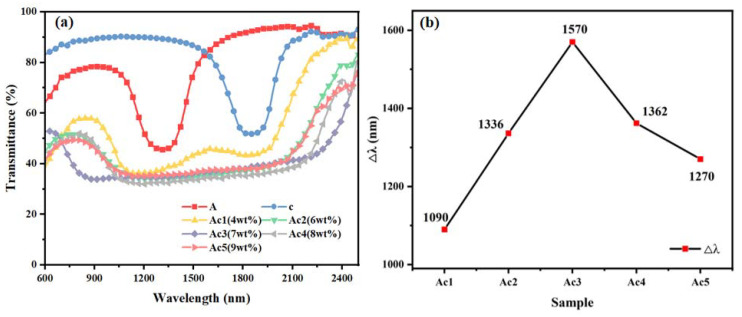
Samples of group Ac: transmission spectra (**a**) and trend of Δλ with the mass fraction (**b**) of polymerizable monomer.

**Figure 3 molecules-28-07063-f003:**
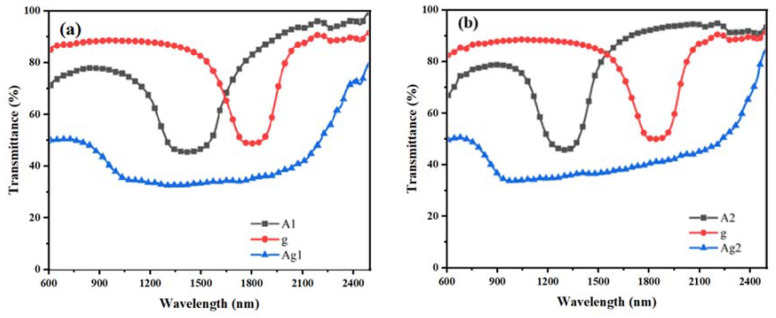
Samples of group Ag: transmission spectra (**a**–**e**) and trend of Δλ with R5011 content (**f**).

**Figure 4 molecules-28-07063-f004:**
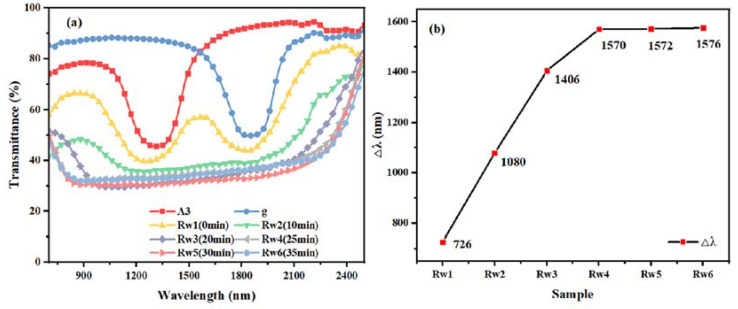
Samples of group Rw: transmission spectra (**a**) and trend of Δλ with polymerization time (**b**).

**Figure 5 molecules-28-07063-f005:**
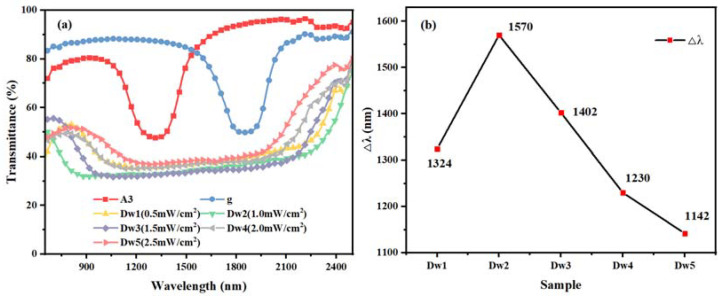
Samples of group Dw: transmission spectra (**a**) and trend of Δλ with UV intensity (**b**).

**Figure 6 molecules-28-07063-f006:**
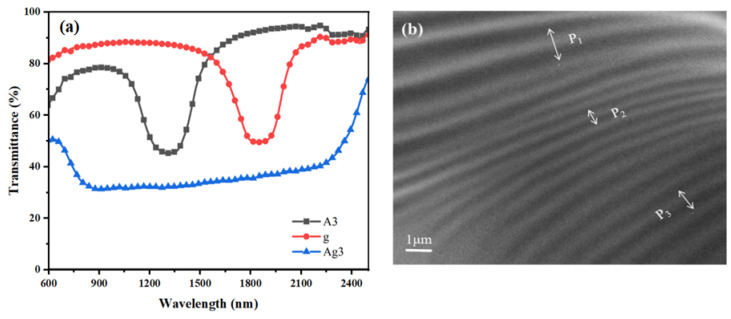
Sample Ag3: transmission spectra before and after polymerization (**a**) and cross-sectional SEM morphology (**b**). The arrows in (**b**) shows the pitch.

**Figure 7 molecules-28-07063-f007:**
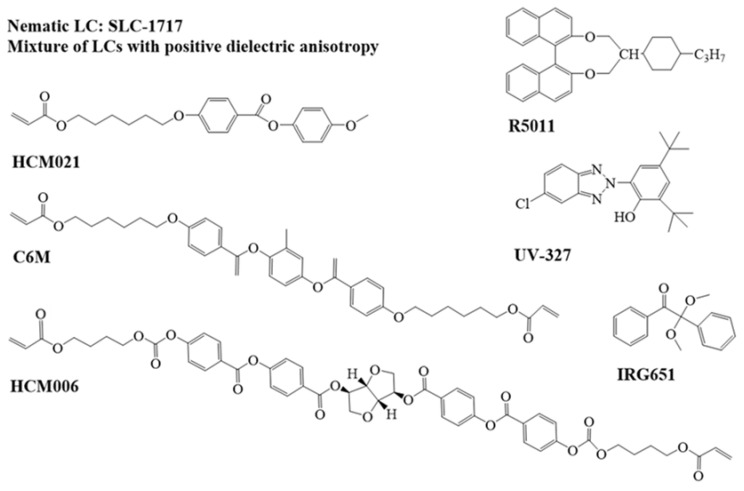
The chemical structure of the experimental materials.

**Figure 8 molecules-28-07063-f008:**
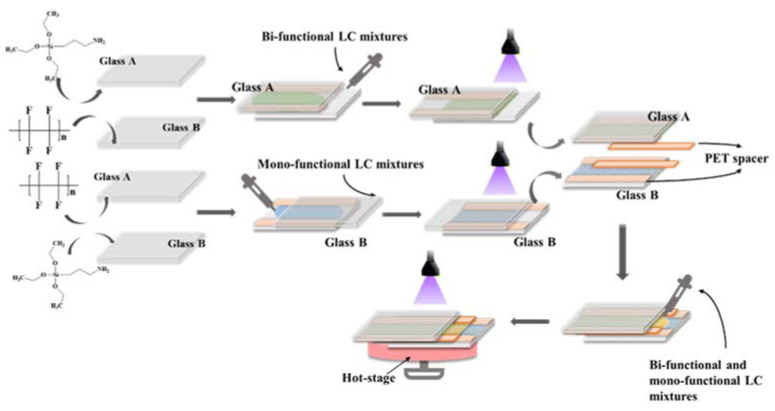
The preparation process of PSCLC films.

**Table 1 molecules-28-07063-t001:** Sample group allocation ratios.

Sample Number	Upper: SLC-1717/C6M/ R5011/IR651/ UV-327 (wt%)	Lower: SLC-1717/HCM021/ HCM006/IR651/ UV-327 (wt%)	Middle: SLC-1717/C6M/ HCM021/IR651 (wt%)
Ag1	90.10/8.00/1.00/0.30/0.60	89.30/8.00/2.10/0.30/0.30	85.70/7.00/7.00/0.30
Ag2	90.05/8.00/1.05/0.30/0.60	89.30/8.00/2.10/0.30/0.30	85.70/7.00/7.00/0.30
Ag3	90.00/8.00/1.10/0.30/0.60	89.30/8.00/2.10/0.30/0.30	85.70/7.00/7.00/0.30
Ag4	89.95/8.00/1.15/0.30/0.60	89.30/8.00/2.10/0.30/0.30	85.70/7.00/7.00/0.30
Ag5	89.90/8.00/1.20/0.30/0.60	89.30/8.00/2.10/0.30/0.30	85.70/7.00/7.00/0.30
Ac1	90.00/8.00/1.10/0.30/0.60	89.30/8.00/2.10/0.30/0.30	91.70/4.00/4.00/0.30
Ac2	90.00/8.00/1.10/0.30/0.60	89.30/8.00/2.10/0.30/0.30	87.70/6.00/6.00/0.30
Ac3	90.00/8.00/1.10/0.30/0.60	89.30/8.00/2.10/0.30/0.30	85.70/7.00/7.00/0.30
Ac4	90.00/8.00/1.10/0.30/0.60	89.30/8.00/2.10/0.30/0.30	83.70/8.00/8.00/0.30
Ac5	90.00/8.00/1.10/0.30/0.60	89.30/8.00/2.10/0.30/0.30	81.70/9.00/9.00/0.30
Rw1	90.00/8.00/1.10/0.30/0.60	89.30/8.00/2.10/0.30/0.30	85.70/7.00/7.00/0.30
Rw2	90.00/8.00/1.10/0.30/0.60	89.30/8.00/2.10/0.30/0.30	85.70/7.00/7.00/0.30
Rw3	90.00/8.00/1.10/0.30/0.60	89.30/8.00/2.10/0.30/0.30	85.70/7.00/7.00/0.30
Rw4	90.00/8.00/1.10/0.30/0.60	89.30/8.00/2.10/0.30/0.30	85.70/7.00/7.00/0.30
Rw5	90.00/8.00/1.10/0.30/0.60	89.30/8.00/2.10/0.30/0.30	85.70/7.00/7.00/0.30
Rw6	90.00/8.00/1.10/0.30/0.60	89.30/8.00/2.10/0.30/0.30	85.70/7.00/7.00/0.30
Dw1	90.00/8.00/1.10/0.30/0.60	89.30/8.00/2.10/0.30/0.30	85.70/7.00/7.00/0.30
Dw2	90.00/8.00/1.10/0.30/0.60	89.30/8.00/2.10/0.30/0.30	85.70/7.00/7.00/0.30
Dw3	90.00/8.00/1.10/0.30/0.60	89.30/8.00/2.10/0.30/0.30	85.70/7.00/7.00/0.30
Dw4	90.00/8.00/1.10/0.30/0.60	89.30/8.00/2.10/0.30/0.30	85.70/7.00/7.00/0.30
Dw5	90.00/8.00/1.10/0.30/0.60	89.30/8.00/2.10/0.30/0.30	85.70/7.00/7.00/0.30

## Data Availability

Data are contained within the article.

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
