# Peer review of "Multilayer, Broadband Infrared Reflectors Based on the Photoinduced Preparation of Cholesteric Liquid Crystal Polymers"

_molecules, 2023, doi:10.3390/molecules28207063_

Round 1

Reviewer 1 Report

The manuscript "Multilayer, broadband infrared reflectors based on photo-induced preparation of cholesteric liquid crystal polymers" by Yutong Liu and co-authors presents a study of ultra-width broadband reflective cholesteric liquid crystal films. A significant gradient of cholesteric helix pitch was achieved by using of three different layers and selection of monomers' polymerization conditions. The manuscript discusses the influence of the monomers and chiral compounds content, UV irradiation intensity and polymerization time on the reflected bandwidths has been studied.

This research is done well and may be interesting to readers of Molecules. The manuscript can be published in Molecules after some minor revisions. In particular:

1. Why did the authors used KH-570 and polytetrafluoroethylene films as orientation agents? How does the choice of an orienting agent affect the formed structure of the liquid crystal and the broadband reflective width?

2. The formation of the helix pitch gradient is explained by the diffusion of molecules, the rate of which depends on various parameters (concentration, temperature, polymerization rate, the molecule chemical structure, etc.). Moreover, several types of molecules are involved in this diffusion process. This makes the presented mechanism of the broadband reflection phenomenon very difficult to understand and perceive. In particular, Subsection 2.1 is very difficult to read and understand. Could the authors provide references to articles where these diffusion processes are studied in more detail, or add a more detailed mathematical description of the diffusion process dynamics to the manuscript?

3. Today, there are several methods for obtaining a cholesteric helix pitch gradient to produce materials with a wide reflection broadband. Could the authors explain the advantage of their method?

4. Is the caption for Figure 7 correct?

Author Response

Dear reviewer,

Thank you very much for your kind attention to consider our work in the manuscript entitled "Multilayer, broadband infrared reflectors based on photo-induced preparation of cholesteric liquid crystal polymers" (molecules-2644044) authored by Yutong Liu, Rui Han, Xiaohui Zhao, Yue Cao, Hui Cao, Yinjie Chen, Zhou Yang, Dong Wang, and Wanli He. In addition, we truly appreciate the reviewers for their careful reading of our manuscript and valuable suggestions. The comments of the reviewers give us a chance to benefit from their wisdom and experience and are very helpful for improving the quality of our manuscript. Then, we have detailedly revised the manuscript according to the comments. A red highlighted version of the manuscript indicating where changes have been made is also included with our submission.

Following is our response to the comments of the Reviewer 1:

  1. The reviewer’s comments: Why did the authors used KH-570 and polytetrafluoroethylene films as orientation agents? How does the choice of an orienting agent affect the formed structure of the liquid crystal and the broadband reflective width?

Author Response: We thank the reviewer for this suggestion. In the study of this manuscript, it is the process of preparing monolayer films containing different compositions and then stacking them. Since the liquid crystal cassettes were used in the film preparation process and both sides of the film were in full contact with the glass, the lipophilic KH-570 and the lipophobic polytetrafluoroethylene were used. In subsequent experiments, the film could be smoothly removed and reassembled into a new liquid crystal cartridge. The use of treated liquid crystal cassettes does not damage the film in the process of removing the film. Orientation was done in the experiment because a more optimal pitch distribution pattern could be formed and the measured broadband reflections were more pronounced, making results more accurate.

  1. The reviewer’s comments: The formation of the helix pitch gradient is explained by the diffusion of molecules, the rate of which depends on various parameters (concentration, temperature, polymerization rate, the molecule chemical structure, etc.). Moreover, several types of molecules are involved in this diffusion process. This makes the presented mechanism of the broadband reflection phenomenon very difficult to understand and perceive. In particular, Subsection 2.1 is very difficult to read and understand. Could the authors provide references to articles where these diffusion processes are studied in more detail, or add a more detailed mathematical description of the diffusion process dynamics to the manuscript?

Author Response: We thank the reviewer for this suggestion. In this study, diffusion of dye is mainly utilized to cause a non-uniform distribution of UV intensity, and polymerization of bis-acrylate monomers occurs after moving to the stronger side of the UV to form an inhomogeneous polymer network, while at the same time, the chiral compounds and the monoacrylate diffuse in the opposite direction. According to diffusion kinetics, ideally substances generally diffuse from the high concentration to the low concentration side. As polymerization proceeds, less unpolymerized bisacrylate monomer remains on the stronger UV side, which promotes diffusion. However, the reflective broadband of the film is not only related to the diffusion rate but also to the polymerization rate. If the polymerization rate is moderate and the molecules are given proper diffusion time, a better state of chiral compound concentration gradient distribution can be formed. If the rate of polymerization is greater than the rate of diffusion, the molecules do not diffuse well, and if the rate of polymerization is less than the rate of diffusion, the molecules diffuse completely and no concentration gradient is formed. Neither of these two cases can form a good pitch gradient. So different influencing factors are discussed in the manuscript.

[Wang, F.; Li, K.; Song, P.; Wu, X.; Chen, H.; Cao, H., The effects of thermally induced diffusion of dye on the broadband reflection performance of cholesteric liquid crystals films. Compos Part B-Eng 2013, 46, 145-150.]

[Li, F.; Wang, L.; Sun, W.; Liu, H.; Liu, X.; Liu, Y.; Yang, H., Dye induced great enhancement of broadband reflection from polymer stabilized cholesteric liquid crystals. Polym Adv Technol 2012, 23 (2), 143-148.]

[Yu, M.; Wang, L.; Nemati, H.; Yang, H.; Bunning, T.; Yang, D.-K., Effects of polymer network on electrically induced reflection band broadening of cholesteric liquid crystals. J Polym Sci Pol Phys 2017, 55 (11), 835-846.]

  1. The reviewer’s comments: Today, there are several methods for obtaining a cholesteric helix pitch gradient to produce materials with a wide reflection broadband. Could the authors explain the advantage of their method?

Author Response: We thank the reviewer for this question. Nowadays, the methods widely used to prepare broadband reflective cholesteric liquid crystal films include: layer stacking method, heat-induced molecular diffusion method, the HTP value change method of chiral compounds, electromagnetic-field-induced pitch non-uniform distribution method, and so on.

  • Huang et al. stacked three layers of ChLC polymers with different pitches to make a composite system with reflectance waveforms covering the reflectance range of each layer. The advantages of the single pitch multilayer stacking method are the simplicity of the preparation process and the controllability of the reflection wavelength center and the reflection wavewidth range. [Huang, Y.; Zhou, Y.; Wu, S.-T., Broadband circular polarizer using stacked chiral polymer films. Express 2007, 15 (10), 6414-6419.]
  • Mitov proposed that liquid crystal oligomer films of cyclosiloxane side chains with different proportions of chiral and achiral side chain branching are directly stacked, and a pitch gradient is formed between the two films by thermal diffusion after certain heat treatments. [Mitov, M., Cholesteric liquid crystals with a broad light reflection band. Adv Mater 2012, 24 (47), 6260-6276.]
  • Yang et al. prepared azo chiral compounds with photoresponsive properties. UV irradiation induced cis-trans isomerization of the azo group, which led to a decrease in the helical twisting force, red-shifting of the reflection peaks, and the cholesteric-phase reflection peaks changed with the increase of UV irradiation time. The cis-trans isomerization of azo chiral compounds at different positions occurs to different degrees, resulting in a gradient distribution of helical twisting force and the significant enhancement of reflectivity while realizing broadband reflection. [Chen, X.; Wang, L.; Chen, Y.; Li, C.; Hou, G.; Liu, X.; Zhang, X.; He, W.; Yang, H., Broadband reflection of polymer-stabilized chiral nematic liquid crystals induced by a chiral azobenzene compound. Chem commun 2014, 50 (6), 691-694.]
  • Yang added an anionic chiral ionic liquid containing chiral groups to the ChLC material system. Under the action of an applied high-frequency AC electric field, the anions in the ionic liquid move toward the positive electrode, resulting in a higher concentration of chiral groups near the positive electrode and a lower concentration near the negative electrode, which creates a gradient of concentrated chiral compounds. Reflected wavelengths cover the entire visible wavelength band when the applied voltage reaches 40V. [Hu, W.; Zhao, H.; Song, L.; Yang, Z.; Cao, H.; Cheng, Z.; Liu, Q.; Yang, H., Electrically Controllable Selective Reflection of Chiral Nematic Liquid Crystal/Chiral Ionic Liquid Composites. Mater. 2010, 22 (4), 468-+.]
  1. The reviewer’s comments: Is the caption for Figure 7 correct?

Author Response: We thank the reviewer for this suggestion. In the new manuscript, we have fixed the errors pointed out by the reviewer.

Thank you and best regards.

Yours sincerely,

Hui Cao

Reviewer 2 Report

Dear Author,

 I would likt to thank for the opportunity to read the manuscript entitled "Multilayer, broadband infrared reflectors based on photoinduced preparation of cholesteric liquid crystal polymers" sento to the journal Molecules, The manuscript is well written and interesting.

I have some comments:

1) All the written parts in the figures are very small. The font size should be increased, especially the legend in Figure 1 and the axis labels.

2) The discussion on the transmission spectra should be extended. Is there a way to simulate the spectra, with analytical or numerical tools? Is there a way to model/predict the shift Delta(lambda)?

Reviewer 3 Report

In this manuscript, a three-layer liquid crystal composite system was proposed to create pitch gradient in PSCLC films.  The results are interesting and data is well represented. However, the authors need to address few things before going to publish.

1.      Can you discuss about stability and reproducibility of the films prepared?

2.      I totally agree, it’s a new technique, if IR reflecting materials are available for IR windows, I see the proposed methods has complications in methodology and too many compounds, discuss its applications and long durability of the system

Author Response

Dear Reviewer,

Thank you very much for your kind attention to consider our work in the manuscript entitled "Multilayer, broadband infrared reflectors based on photo-induced preparation of cholesteric liquid crystal polymers" (molecules-2644044) authored by Yutong Liu, Rui Han, Xiaohui Zhao, Yue Cao, Hui Cao, Yinjie Chen, Zhou Yang, Dong Wang, and Wanli He. In addition, we truly appreciate the reviewers for their careful reading of our manuscript and valuable suggestions. The comments of the reviewers give us a chance to benefit from their wisdom and experience and are very helpful for improving the quality of our manuscript. Then, we have detailedly revised the manuscript according to the comments. A red highlighted version of the manuscript indicating where changes have been made is also included with our submission.

Following is our response to the comments of the Reviewer 3:

  1. The reviewer’s comments:Can you discuss about stability and reproducibility of the films prepared?

Author Response: We thank the reviewer for this suggestion. First, we think the film is more stable. The pitch gradient for the formation of the film is anchored by the polymer network formed by polymerization, and once formed, other molecules that can make diffusive motion are impeded in their motion by the polymer network, thus maintaining the result unchanged, cf. Subsection 2.4 in the manuscript, with all other variables held constant, the polymerization time is sufficient and the reflective broadband of the film will no longer grow after the acrylate is fully polymerized. Polymer networks formed by cross-linking of acrylates have been extensively studied in the past for their good chemical stability and physical strength.

Next, we think the film is also reproducible. In the course of our study, we firstly discussed the effects of various factors on the reflective bandwidth of the films separately, determined the optimal experimental conditions in each group respectively, and then finally repeated the experimental process according to the optimal experimental conditions, obtaining films with good experimental results (e.g., Subsection 2.6).

  1. The reviewer’s comments: I totally agree, it’s a new technique, if IR reflecting materials are available for IR windows, I see the proposed methods has complications in methodology and too many compounds, discuss its applications and long durability of the system.

Author Response: Thanks for the reviewer’s suggestion. The findings in this manuscript are mainly in the near-infrared region. In real life, the primary application is to attach this film to the windows of the buildings to obtain infrared adjustable smart windows, which can effectively adjust the incidence and reflection of infrared light, and can effectively reduce the indoor temperature in the summer, reduce the frequency of using air conditioners, and realize energy saving and emission reduction. It can also be used in military and medical applications as the laser protection film. The wider the broadband of the shielding, the better the protection film for medical and military applications. As for durability, on the same basis as mentioned above, the polymer network formed by cross-linking of acrylates has good chemical stability and physical strength, enabling the film to achieve long-time use.

[Ranjkesh, A.; Choi, Y.; Huh, J.-W.; Oh, S.-W.; Yoon, T.-H., Flexible, broadband, super-reflective infrared reflector based on cholesteric liquid crystal polymer. Sol Energ Mat Sol C 2021, 230, 111137.]

[Zhang, L.; Wang, M.; Wang, L.; Yang, D.-K.; Yu, H.; Yang, H., Polymeric infrared reflective thin films with ultra-broad bandwidth. Liq Cryst 2016, 43 (6), 750-757.]

Thank you and best regards.

Yours sincerely,

Hui Cao

Reviewer 4 Report

The study proposed by Liu and coworkers explores a polymer network of polymer-stabilized cholesteric liquid crystal (PSCLC) with a gradient pitch for the realization of a broadband reflection of PSCLC films. The results are very detailed and interesting and will surely impact the subsequent experiments and engineering applications for the LC community. I found the manuscript well-organized and well-documented. The materials and methods are correctly described. I consider it suitable for publication.

The manuscript would benefit from a thorough proofreading to correct some of the grammar. 

Author Response

Dear Reviewer,

Thank you very much for your kind attention to consider our work in the manuscript entitled "Multilayer, broadband infrared reflectors based on photo-induced preparation of cholesteric liquid crystal polymers" (molecules-2644044) authored by Yutong Liu, Rui Han, Xiaohui Zhao, Yue Cao, Hui Cao, Yinjie Chen, Zhou Yang, Dong Wang, and Wanli He. In addition, we truly appreciate the reviewers for their careful reading of our manuscript and valuable suggestions. The comments of the reviewers give us a chance to benefit from their wisdom and experience and are very helpful for improving the quality of our manuscript. Then, we have detailedly revised the manuscript according to the comments. A red highlighted version of the manuscript indicating where changes have been made is also included with our submission.

Following is our response to the comments of the Reviewer 4:

  1. The reviewer’s comments: The manuscript would benefit from a thorough proofreading to correct some of the grammar.

Author Response: We thank the reviewer for this suggestion. In the new manuscript, we have fixed the errors pointed out by the reviewer.

Thank you and best regards.

Yours sincerely,

Hui Cao

Round 2

Reviewer 2 Report

The authors have properlt worked to submit a response to my comments and the manuscript has been accordingly revised. I would suggest to the editorial borad to accept the manuscript for publication.